# Community Wastewater-Based Surveillance Can Be a Cost-Effective Approach to Track COVID-19 Outbreak in Low-Resource Settings: Feasibility Assessment for Ethiopia Context

**DOI:** 10.3390/ijerph19148515

**Published:** 2022-07-12

**Authors:** Solomon Ali, Esayas Kebede Gudina, Addisu Gize, Abde Aliy, Birhanemeskel Tegene Adankie, Wondwossen Tsegaye, Gadissa Bedada Hundie, Mahteme Bekele Muleta, Tesfaye Rufael Chibssa, Rediet Belaineh, Demessa Negessu, Dereje Shegu, Guenter Froeschl, Andreas Wieser

**Affiliations:** 1Department of Microbiology, Immunology and Parasitology, St. Paul’s Hospital Millennium Medical College, Addis Ababa P.O. Box 1271, Ethiopia; addisu.gize@sphmmc.edu.et (A.G.); meskelbirehane@gmail.com (B.T.A.); wondewosentsg@gmail.com (W.T.); gadissa.bedada@sphmmc.edu.et (G.B.H.); 2Department of Internal Medicine, Jimma University, Jimma P.O. Box 378, Ethiopia; esayas.gudina@ju.edu.et; 3National Animal Health Diagnostic and Investigation Center, Sebeta P.O. Box 04, Ethiopia; abde.aliy@yahoo.com (A.A.); chibssasafo@gmail.com (T.R.C.); red.tkbr@yahoo.com (R.B.); dnegessu@gmail.com (D.N.); dshegu@yahoo.com (D.S.); 4Department of Surgery, St. Paul’s Hospital Millennium, Medical College, Addis Ababa P.O. Box 1271, Ethiopia; mahteme.bekele@sphmmc.edu.et; 5Division of Infectious Diseases and Tropical Medicine, University Hospital, Ludwig-Maximilians-Universität, 80802 Munich, Germany; guenter.froeschl@med.uni-muenchen.de (G.F.); wieser@mvp.lmu.de (A.W.); 6Partner Site Munich, German Center for Infection Research (DZIF), 80802 Munich, Germany; 7Max von Pettenkofer-Institute, Ludwig-Maximilians-Universität, 80802 Munich, Germany

**Keywords:** wastewater, wastewater-based epidemiology, MBR, COVID-19, SARS-CoV-2, RT-PCR, SPHMMC

## Abstract

Wastewater surveillance systems have become an important component of COVID-19 outbreak monitoring in high-income settings. However, its use in most low-income settings has not been well-studied. This study assessed the feasibility and utility of wastewater surveillance system to monitor SARS-CoV-2 RNA in Addis Ababa, Ethiopia. The study was conducted at nine Membrane Bio-reactor (MBR) wastewater processing plants. The samples were collected in two separate time series. Wastewater samples and known leftover RT-PCR tested nasopharyngeal swabs were processed using two extraction protocols with different sample conditions. SARS-CoV-2 wastewater RT-PCR testing was conducted using RIDA GENE SARS-CoV-2 RUO protocol for wastewater SARS-CoV-2 RNA testing. Wastewater SARS-CoV-2 RNA RT-PCR protocol adaptation, optimization, and detection were conducted in an Addis Ababa, Ethiopia context. Samples collected during the first time series, when the national COVID-19 case load was low, were all negative. Conversely, samples collected during the second time series were all positive, coinciding with the highest daily reported new cases of COVID-19 in Ethiopia. The wastewater-based SARS-CoV-2 surveillance approach is feasible for Addis Ababa. The COVID-19 wastewater based epidemiological approach can potentially fill the evidence gap in distribution and dynamics of COVID-19 in Ethiopia and other low-income settings.

## 1. Introduction

Despite the remarkable worldwide advancements in capability for early case detection, implementation of COVID-19 vaccination programs, and extensive invigoration of non-pharmacological preventive measures [1], the COVID-19 global pandemic is ongoing, and in many countries, the incidence of infections is on the rise again. Considering this fact, some scholars are speculating the potential coexistence of the world with COVID-19 disease as a new normal [2].

As of 29 June 2022, an estimate of the Johns Hopkins University depicts globally more than half a billion cumulative cases of COVID-19, more than 6 million deaths, and at the same time, more than 11 billion administered vaccine doses [3,4]. Ethiopia performed a cumulative of around 5 million COVID-19 tests as of 29 June 2022, corresponding with less than 5% of its total population. During the same period, the country reported 488,108 confirmed cases, 7535 COVID-19-related deaths, and about 42 million administered doses of the vaccine [3,5]. Due to a presumably low detection rate, the actual incidence of COVID-19 cases in Ethiopia may be much higher than the reported one. A recent sero-epidemiological survey among health care workers and communities in Addis Ababa and Jimma has indicated a sero-prevalence of more than 70%, indicating that cases remain largely undetected [6]. More than half of these cases and about 62% of COVID-19-related deaths in Ethiopia were reported from Addis Ababa, home to less than 5% of the national population [7]. Besides the understandable high burden of the cases and deaths in urban metropolitans, the observed COVID-19 burden in the capital may indicate a testing bias towards large cities with relatively better access to testing infrastructure. The true epidemiological picture of the pandemic is thus likely to be greatly different from what has officially been reported, as could be demonstrated by serological studies [6].

During early stages of the pandemic, the global COVID-19 control strategy has focused on non-pharmacological measures mainly targeting individuals [8]. Compliance to these recommended public health measures is highly dependent on the individual level of understanding, willingness, behavior, and available resources. In addition, epidemiological data, which has been used to estimate virus circulation, transmission and incidence, has been found to be highly dependent on the currently applied testing strategies and measures such as lock-downs or the closure of schools. The search for alternatives for estimations of virus circulation has brought about the investigation of urban wastewater, which allows an early detection of virus circulation in a given catchment area, and even a geographic mapping of transmission, based on the sampling at different wastewater collection points [9,10,11,12,13,14,15,16,17,18,19,20,21].

This COVID-19 wastewater-based epidemiology approach has created an opportunity to understand the distribution and dynamics of the disease in defined sewage catchment areas. The COVID-19 mitigation strategies can therefore be differentially targeted to communities where sewage samples can indicate outbreaks early on in a given catchment area. However, most of the published data on wastewater epidemiology was generated from higher income settings, such as Europe, Australia, the United States (US), China, India, Japan, and South Africa, the only country from Africa [9,10,11,12,13,14,15,16,17,18,19,20,21]. Arguably, evidence generated from high-income countries might not be directly applicable for low-resource settings due to differences in geo-climatic and cultural aspects, wastewater composition, and the applied technologies in the wastewater management system.

Wastewater-based epidemiology of infectious diseases is not a new concept. It has been utilized for enteric viruses as an early warning system [22]. With the emergence of the COVID-19 pandemic, it has now received large global attention for its simplicity and cost effectiveness [10,11,12,13,14,15,16,17,18,19,20,21,22]. Apart from the manuscripts mentioned above, there are also many articles including reviews addressing critical issues related to waste management, pollution, policy and regulations in China, Indonesia, Bangladesh, India and Malaysia during the era of the COVID-19 pandemic [23,24,25,26].

We believe that the capability for detection and monitoring of SARS-CoV-2 RNA from wastewater systems in low- and middle-income-countries (LMICs) such as Ethiopia would not only mend the evidence gaps about the true burden of COVID-19, but it would also generate the capacity to monitor other infectious diseases with enteric excretion in the future.

Thus, this study aims to assess the feasibility of the approach, build a wastewater-based epidemiology capacity, and to give first insights into data generated for the assessment and monitoring of SARS-CoV-2 RNA from wastewater management plants in Addis Ababa, Ethiopia.

## 2. Materials and Methods

### 2.1. Study Setting

Addis Ababa is the capital city of Ethiopia, with an estimated population of 5.2 million as of 2022 [27]. Administratively, it comprises 11 sub-cities. There are two central wastewater treatment ponds that are connected by a sewage network, serving a catchment population of about 10% of the total urban population. In addition, there are 14 decentralized wastewater processing plants at different condominium sites [28]. Addis Ababa Water and Sewerage Authority (AAWSA) is the governing body of the municipal wastewater system.

The rapid booming of the city with common housing projects (mostly condominium construction) in the past 15 years forced AAWSA to plant containerized Membrane Bioreactors as wastewater treatment plants. These plants are designed to transform dissolved and particulate constituents into less hazardous end products by combining conventional activated sludge processes with membrane separation. The principle and detailed operations of MBR units is described elsewhere [29].

The MBR units in Addis Ababa have been serving the wastewater processing need of fourteen residential complexes. The units have a treatment capacity of 20,000 cubic meter of wastewater per day [29]. For this study, we included nine MBR units representing 64% of the total units planted in Addis Ababa (Figure 1).

### 2.2. Waste Water Sample Collection

Wastewater samples from participating MBR units were collected longitudinally and in two separate time series using an integrated sampling method. In the first series, the inlet wastewater samples were collected during a period of 25 October 2020 to 13 December 2020 every Sunday morning between 7:00 AM and 9:00 AM. The once weekly rhythm at the same time period was chosen in order to reduce time dependent variations in the use of water- and sewage-infrastructure by the population. On Sunday mornings in particular, most inhabitants in Addis Ababa are at home; thus, the sewage collected from residential areas is dominated by the wastewater generated in the private toilets. The average distance from private toilet to MBR units is three km, resulting in short flow times from the sink to the treatment plant. Accordingly, the sewage is not hermetically closed; thus, there is a possibility of wastewater to be mixed with rain water in sewage lines on the way to the MBR. The samples (300 mL) were collected as qualified spot samples in duplicates and transported to St. Paul’s Hospital Millennium Medical College (SPHMMC) using an ice-loaded cool box. At SPHMMC, the samples were transferred to −80 °C freezers immediately and later brought to the National Animal Health and Diagnostic Investigation Center (NAHDIC).

In the second time series, the wastewater samples collection was resumed for three weekly time points from 25 December 2021 to 9 January 2022 to acquire some information about the epidemiology of the fourth COVID-19 wave in Ethiopia and compare the inlet and aeration samples for viral nucleic acid recovery. The samples were collected using cool boxes with ice packs and processed freshly within 24 h at NAHDIC.

The NAHDIC is the referral and reference veterinary medicine laboratory in Ethiopia. It is located in Sebeta, 25 km southwest of Addis Ababa. It is the center of excellence for animal disease surveillance, investigation, diagnosis, and research. NAHDIC has implemented an ISO/IEC 1725 quality system [30]. It should be noted here that there was no sample collection for eleven months between the two time series data collection points. This is due to the fact that the main goal of this study is to assess the feasibility of a COVID-19 wastewater-based epidemiology in an Ethiopian (low resource settings) context. Despite the resource constraints, we did our best to collect data that covers low and high COVID-19 community transmission time points to generate representative data. Furthermore, it is not the intent of this study to understand the dynamics of SARS-CoV-2 RNA concentration in wastewater throughout the two year period of the COVID-19 pandemic in Ethiopia. Thus, to balance the limited resources we have while generating scientifically valid representative data, the time points between January–November 2021 were not included.

#### Wastewater Sample Processing and RNA Concentration

Fresh wastewater samples were processed within 24 h of collection. Frozen samples were thawed on ice and subsequently processed. First, the samples were sieved using gauze to separate sludge from liquid. 50 mL of sieved wastewater samples were transferred to Corning falcon tubes and centrifuged at 2500× *g* for 20 min at 4 °C using a Thermo Scientific JOUAN CR4i centrifuge [9]. Thirty-eight mL of the supernatant was transferred to ultracentrifuge tubes and placed in a Beckman Coulter Avanti JXN-30 ultracentrifuge using the JA-25.50 rotor. The tubes were centrifuged at a speed of 27,000× *g* at 4 °C for one hour [9]. The supernatants were discarded carefully using a 30 mL automatic biuret. The pellets were re-suspended in 500 μL nuclease free water provided in the QIAGEN kit and immediately transferred to the molecular laboratory for RNA extraction.

### 2.3. RNA Extraction 

RNA extraction was executed using an Allprep powerVial DNA/RNA extraction kit (QIAGEN, Hilden, Germany) and QIAamp Viral RNA Mini kit (QIAGEN, Hilden, Germany) following the manufacturer’s instruction [31,32].

### 2.4. Master Mix and RT-PCR Test

The master mix was prepared following BGI and RIDA GENE SARS-CoV-2 RUO (r-biopharm, Darmstadt, Germany) RT-PCR testing protocols depending on the experiment types. As per BGI (Shenzhen, China) protocol [33], a master mix for a single RT-PCR reaction was prepared by mixing 18.5 μL of reaction mix (including reagent for amplification, probes, primers targeted SARS-CoV-2 ORF1ab gene, ICR) with 1.5 μL of enzyme mix [33]. The 20 μL of master mix was dispensed in each reaction well including the no template (negative) control and the positive control (standard). Finally, 10 μL of no template control, wastewater eluate and positive control were added in the respective designated reaction wells [33]. The reaction wells were sealed and briefly centrifuged before they were placed for amplification into an ABI 7500 RT-PCR system. The ABI 7500 RT-PCR was programmed to run one cycle at 50 °C for 20 min, one cycle at 95 °C for 10 min, for 40 cycles (95 °C for 15 s and 60 °C for 30 s) [33].

RIDA GENE SARS-CoV-2 RUO (r-biopharm, Darmstadt, Germany) master mix was prepared by mixing 19.3 μL of reaction mix (including primer for target E-gene), 0.7 μL of Taq-Polymerase, and 1 μL of ICR [34]. Twenty-one μL of the master mix was dispensed in each reaction wells including no template and positive control wells. Five μL of each no template control, wastewater eluate, and positive control was dispensed in each respective reaction well with master mix [34].

The micro-well plate was sealed and loaded on Applied Biosystems Real-Time PCR Instruments (ABI 7500) with the following PCR profile; Reverse transcription 10 min, 58 °C, initial denaturation 1 min, 95 °C, cycles 45 (PCR denaturation 15 s, 95 °C, annealing/extension 30 s, 60 °C), and temperature transition rate/ramp rate: maximum.

### 2.5. RT-PCR Signal Detection

The detection channel was set as Fluorescein amides (FAM) for SARS-CoV-2 RNA and Victoria (VIC) for ICR (Internal Control Reaction) for both BGI and RIDA master mix tests, as suggested by the manufacturer of the assays. The auto cycle threshold (Ct) and baseline functions of the ABI 7500 Fast Real-Time PCR System software version 1.4, Singapore were used to analyze the data.

### 2.6. Quality Control and Interpretation of the PCR Result

All RT-PCR readings of this study were quality assured and interpreted using the criteria described in Table 1. The contents of the table are summarized from the RT-PCR testing result interpretation recommendations of the BGI (Shenzhen, Guangdong, China) and r-biopharm (Darmstadt, Germany) [33,34] RT-PCR testing protocols.

## 3. Results

### 3.1. Optimization of SARS-CoV-2 Wastewater RT-PCR Testing

Adaptation, customization, and optimization of wastewater SARS-CoV-2 RNA RT-PCR testing to the local context was conducted. For this purpose, a total of six different trials were performed using two different extractions and SARS-CoV-2 RNA RT-PCR testing protocols at different sampling conditions. The experiment conditions and observed results were presented in Table 2. Detailed description of the experiments was attached as an appendix at the end of the manuscript (Appendix A).

### 3.2. Stored and Fresh Wastewater Processing Result

The 72 inlet wastewater samples collected from nine MBR processing plants were RT-PCR tested. The samples were collected over 8 weeks from 25 October 2020 to 13 December 2020 and stored in −80 °C freezer. SARS-CoV-2 RNA was not detected from any of these stored wastewater samples. To give more insight, we present the number of daily COVID-19 new cases and the positivity rate during 8 weeks of this research data collection period (Figure 2). Between October–December 2020, the average daily COVID 19 new cases and positivity rate was 478 and 9%, respectively (Figure 2).

Most (88.9%) of the inlet fresh wastewater samples collected from 25 December 2021 to 9 January 2022 and processed within 24 h of collection were positive for SARS-CoV-2 RNA (Table 3). Furthermore, the positivity was maintained for three consecutive weeks. According to Ethiopian Ministry of Health data, the average number of daily new COVID-19 cases and positivity rates during this second wastewater collection time period have increased to 2874 and 27%, respectively (Figure 2). On the other hand, all fresh wastewater samples collected from the aeration tank during the aforementioned data collection dates at similar conditions were negative for SARS-CoV-2 RNA (Table 3).

## 4. Discussion

In this study, an optimization of the COVID-19 wastewater PCR protocol to fit with the local context was performed successfully. The major challenge during optimization was inhibition. The identification of the types of inhibitors present in the studied Ethiopian wastewater system was beyond the scope of this study. However, wastewater PCR inhibition is usually associated with presence of debris, fulmic acids, metal ions, polyphenol, and high activity of RNase enzymes [35,36]. These chemical compounds can interfere with PCR testing through different mechanisms including degradation of the target nucleic acids [36]. Nonetheless, our finding indicates that wastewater-based surveillance can be used to monitor infectious disease outbreak in settings where the traditional disease surveillance system is difficult due to limited resources for laboratory testing.

All wastewater samples collected at an early stage of the pandemic and stored in −80 °C freezers for more than one year were negative. We can only speculate about the reasons. First of all, during the early stage of the pandemic in Ethiopia, the concentration of SARS-CoV-2 RNA in wastewater might have been just simply below the detection limit of PCR testing. This is directly linked with the number of COVID-19 infected individuals shedding the virus in the sewage catchment area during the data collection period. At that time, the maximum numbers of cases detected weekly in Ethiopia were 4206 [5]. Given this, the Ministry of Health report includes all cases detected in the country; the number of cases detected from Addis Ababa only was obviously lower than the reported weekly number. Thus, the chance of detecting SARS-CoV-2 RNA from wastewater is very low. The usage of gauze to filter the sludge and slow thawing of the stored samples may also contribute for the loss of viral particles or RNA. In addition, the inherent risk of RNA degradation during storage especially in low resource settings with repeated power interruption could not be ignored, even though we are not aware of any such incidence with regard to our used storage facility.

In this study, we have detected SARS-CoV-2 RNA for three consecutive weeks during the second time series of data collection. At this time period of data collection, the number of weekly COVID-19 cases detected from Ethiopia was more than three times higher than the first time series cases (Figure 2). According to Minster of Health data, the highest ever number of weekly detected case in the country was reported between 26 December 2021 and 1 January 2022. This coincided with the second week of our second wastewater data collection period. During this week, a total of 28,590 weekly cases were detected from Ethiopia [5]. Considering 55% of the total detected cases in Ethiopia are from Addis Ababa [7], a calculated estimate of 15,725 cases were reported from Addis Ababa. Recent published evidence estimated that the minimum number of SARS-CoV-2 infected cases needed to detect the viral RNA in wastewater ranged from a 253 to 409/10,000 population [37]. However, the highest 15,725 cases detected from Addis Ababa is only equivalent to 30 cases per 10,000 population considering the population size of Addis Ababa as 5.2 million [27]. This rate is at least eight times lower than the minimum SARS-CoV-2 detection threshold limit reported previously [37]. This data indicates that there are a significant number of COVID-19 cases in the community without being detected by implemented testing strategy of the country. Determining the minimum number of COVID-19-infected cases expected in the community to detect SARS-CoV-2 RNA from wastewater management plants in Addis Ababa was not in the scope of this study. However, generating such evidence would help to forecast the minimum number of SARS-CoV-2 infections in the community.

Several studies have reported that COVID-19 wastewater-based epidemiological approaches provide indirect information about the burden of COVID-19 in a defined community [10,11,12,13,14,15,16,17,18,19,20]. The approach provides a proxy indicator for the concentration of RNA shed by infected individuals at different stages of the disease, including even pre-symptomatic and asymptomatic cases [38]. Accordingly, an increase or a decrease in wastewater viral concentration can be inferred to the number of new or resolved SARS-CoV-2 infections in the community. This may allow tailoring prevention strategies to fit the local context considering chronological dynamics and spatial distribution. In addition, the periodic wastewater monitoring of COVID-19 supplemented with SARS-CoV-2 RNA sequencing data at defined communities allows for the timely detection of the emergence or surge of variants of concern before their detection in direct patient samples.

In resource-limited settings, the capacity to detect and monitor SARS-CoV-2 in wastewater gives a better insight about the distribution and dynamics of the pandemic at lower cost. Due to resource and capacity limitations, the reported number of conducted COVID-19 tests per 1000 population has been extremely low in Ethiopia. To substantiate this with figures from comparable settings, the current estimates (as of 23 June 2022) of the COVID-19 testing density for South Sudan, Ethiopia, Uganda, and Kenya are 36.05, 42.63, 58.34, and 67.80 per 1000 population, respectively. Just for comparison, the current (as of 23 June 2022) estimates of the COVID-19 testing rate for South Africa, Italy, UK, USA and Germany are 427.00, 3725.12, 7371.74, 2741.00, and 1560.04 per 1000 population, respectively [4]. Thus, the effective utilization of the innovative COVID-19 wastewater-based epidemiological approach may minimize the observed huge COVID-19-related evidence gap between high- and low-income-countries without compromising the limited resources in low- and middle-income-countries.

In the Ethiopian context, the COVID-19 wastewater-based epidemiological approach is more feasible in urban settings, especially in metropolitan cities such as the capital, Addis Ababa. In this city, there are 14 Membrane Bioreactor (MBR) units (Figure 1). It is estimated that a single MBR unit is capable of managing the wastewater produced by 15,000 residents or more [29]. Considering this fact, wastewater SARS-CoV-2 monitoring of the 14 MBR units in Addis Ababa could provide indirect epidemiological information about at least 210,000 residents. The condominium sites are separate compounds equipped with dedicated MBR. This makes the setting more convenient for early warning systems and localized implementation of mitigation strategies in cost efficient way to substantiate the magnitude of cost effectiveness, and to give a more detailed insight, we have analyzed recent publication from Ethiopia. The unit cost of detecting one RT-PCR positive COVID-19 case and the unit cost to detect one RT-PCR positive COVID-19 case through contact tracing is stated to be USD 37.70 and USD 54.00, respectively [39]. The same study has indicated that the cost for COVID-19 RT-PCR testing was around USD 3.91 and an additional USD 1.31 was needed for sample collection [39]. In this study, we observed that the cost to test SARS-CoV-2 RNA from one wastewater processing plant by RT-PCR was around USD 300.00, including all consumables. Considering the 14 MBR wastewater processing plants in Addis Ababa serving 210,000 residents, an estimated USD 1,100,400.00 (210,000 × USD 5.24) would be needed to do a onetime COVID-19 RT-PCR community mass testing. Comparatively, the cost of wastewater SARS-CoV-2 wastewater RT-PCR testing from 14 MBR plants in Addis Ababa would only be USD 4200.00. Nonetheless, rural areas and most other towns in Ethiopia to date are generally not served by such wastewater management systems. As a consequence, this system cannot be implemented in most parts of the country. As outbreaks such as COVID-19 frequently affect densely populated urban settings, and due to the fact that an increasing share of the population in LMICs does indeed reside in urban settings, this system remains of paramount relevance to Ethiopia and similar settings.

The entire set of samples collected from the aeration reactor tank was negative for SARS-CoV-2 RT-PCR testing. This finding indicates that the viral RNA could be suffering from degradation during the aeration process. The purpose of wastewater aeration process is to augment microbial growth to allow for the aerobic biodegradation of organic materials [29]. It normally takes 3–4 days at room temperature with continuous air (oxygen) supply. Considering this, the harsh aeration reactor condition coupled with high average atmospheric temperatures (>25 °C) can potentially degrade any RNA target.

This study has some limitations. As resource limitations restricted us to only three times weekly point observations of SARS-CoV-2 RNA, we could not reasonably estimate the weekly wastewater SARS-CoV-2 RNA viral loads and apply statistical tests to estimate the strength of association with weekly RT-PCR-detected cases.

## 5. Conclusions

The detection of SARS-CoV-2 RNA from municipal wastewater in resource-limited countries such as Ethiopia will enable one to survey and monitor the COVID-19 pandemic at a low cost. The COVID-19 wastewater-based epidemiological approach is well applicable to a metropolitan setting such as Addis Ababa, where new residential sites are equipped with dedicated MBR plants. In these settings, wastewater is much more concentrated and polluted with bleach solution and detergents than in areas which offer frequent running water, such as in mid- and high-income settings. The COVID-19 wastewater-based epidemiological approach can potentially fill the evidence gap in the distribution and dynamics of COVID-19 in Addis Ababa. This capacity should be utilized in Ethiopia and elsewhere in LMICs for an evidence-based policy or interventional decisions. The wastewater sample should be collected at the inflow or sewage pipes with short sink-to sample times. Due to large amounts of inhibitory substances in the sewage, good extraction conditions with effective inhibitor removal are required to obtain proper results.

## Figures and Tables

**Figure 1 ijerph-19-08515-f001:**
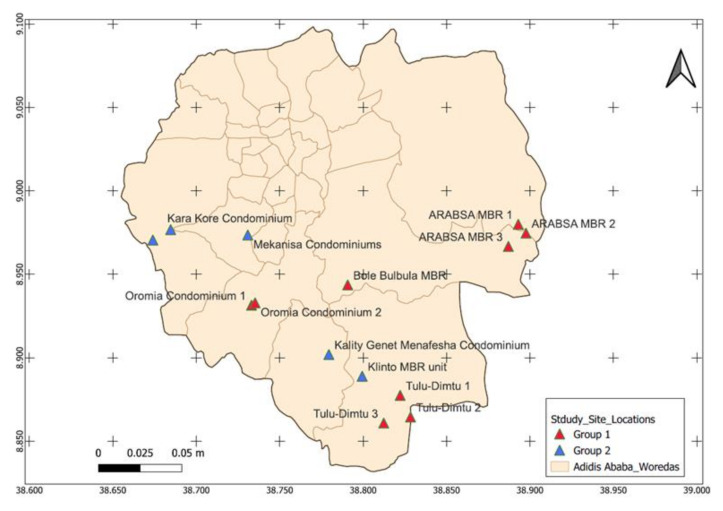
Map of decentralized MBR units in Addis Ababa, where group 1 represents MBR units included in this study and group 2 represents MBR units not included in the current study. Woredas are geographical units in Addis Ababa. Source: this map is created by Chaile Mulu (Geospatial Epidemiologist at National Data Management Center, Ethiopian Public Health Institute).

**Figure 2 ijerph-19-08515-f002:**
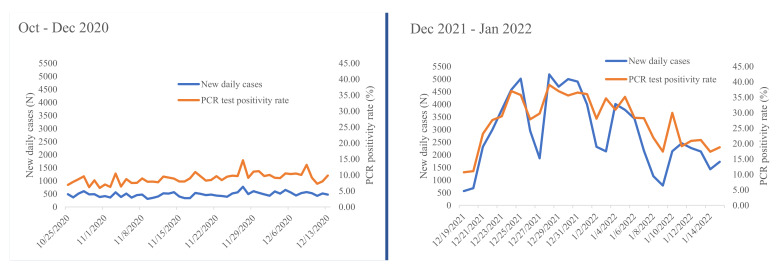
Daily new COVID-19 cases and RT-PCR test positive rate of Ethiopia as reported by the Ministry of Health during our study periods of 19 October to 13 December 2020 and 19 December 2021 to 15 January 2022 (data source: Ministry of Health-Ethiopia daily COVID-19 testing period).

**Table 1 ijerph-19-08515-t001:** Quality control indicators and interpretation of the PCR readings.

QA Metrics	VIC	FAM	Interpretation
**BGI-RT-PCR Kit for Detecting SARS-CoV-2**			
NTC/Blank control	Ct value is 0 or no data available	Ct vale is 0 or no data available	Pass
Standard	S shape amplification curve with Ct ≤ 32	S shape amplification curve with Ct ≤ 32	
Positive sample	S shape amplification curve with Ct ≤ 32	S shape amplification curve with Ct ≤ 38	Valid positive
Negative sample	S shape amplification curve with Ct ≤ 32	Amplification curve not S-shaped with Ct of 0 or No data available	Valid negative
Sample	S shape amplification curve with Ct ≤ 32	S-shape standard curve with Ct > 38	Invalid/inhibited
Sample	Amplification curve with Ct > 32	Amplification curve not S shape with Ct value as 0 or no data available	Invalid/inhibited
**RIDA GENE SARS-CoV-2 RUO Test**			
NTC	Amplification signal with Ct > 20	No Amplification signal with Ct 0	Pass
Standard/Positive control	Amplification signal may or may not be detected.	Amplification signal with Ct range between 25–31	Pass
Positive sample	Amplification signal may or may not be detected.	Amplification signal with Ct < 42	SARS-CoV-2 detectable
Negative sample	Amplification signal with Ct > 20	No amplification signal or Ct > 42	Target gene not detectable
Negative sample	No amplification signal	No amplification signal	Invalid

where QA: quality assurance, Ct: cycle threshold, NTC: no template control, FAM: Fluorescein amides, VIC: Victoria.

**Table 2 ijerph-19-08515-t002:** Detailed descriptions of RNA extraction and PCR testing performed to optimize wastewater COVID-19 RNA detection in Addis Ababa.

Trials	Sample	FAM Detector (Ct)	VIC Detector Ct)	PCR Testing	Interpretation
I	The 12 stored wastewater samples	Undetected	Undetected	* RIDA	Fail
	No template control	Undetected	Undetected		
	Positive control	30	Undetected		
	The 12 stored wastewater samples	Undetected	Undetected	* BGI	Fail
	No template control	Undetected	Undetected		
	Positive control	29	Undetected		
II	The 12 stored wastewater samples	Undetected	Undetected	* RIDA (Repeat)	Fail
	No template control	Undetected	Undetected		
	Positive control	29	Undetected		
III	The 5 stored wastewater samples	Undetected	Undetected	* RIDA Master mix	Fail and invalid test result
	Known PCR negative swab sample	Undetected	Undetected		
	Known PCR positive swab sample	29	Undetected		
	Known PCR positive swab eluate I	25	Undetected		
	Known PCR positive swab eluate II	24	Undetected		
	Known PCR positive swab eluate III	36	Undetected		
	No template control	Undetected	Undetected		
	Positive control	31	Undetected		
IV	The 5 stored wastewater samples	Undetected	Undetected	* BGI master mix	Pass and valid test result for eluate II and III
	Known PCR negative swab sample	Undetected	Undetected		
	Known PCR positive swab sample	33	31		
	Known PCR positive eluate I	29	37		
	Known PCR positive eluate II	27	32		
	Known PCR positive eluate III	39	28		
	No template control	Undetected	Undetected		
	Positive control	31	33		
V	Stored wastewater sample I	Undetected	30	** RIDA	Pass and valid test result
	Stored wastewater sample II	Undetected	31		
	Stored wastewater sample III	Undetected	31		
	Stored wastewater sample IV	Undetected	32		
	Stored wastewater sample V	Undetected	30		
	Stored wastewater sample VI	Undetected	30		
	Stored wastewater sample VII	44.7	31		
	Stored wastewater sample VIII	Undetected	31		
	Known PCR positive Patient swab sample	29.85	31		
	Known PCR negative patient swab sample	Undetected	34		
	No template control	Undetected	30		
	Positive control	28	28		
VI	1:10 diluted + ve control	38.12	Undetected	*** RIDA	Fail and invalid test result
	The 100 μL each wastewater + known PCR positive	Undetected	Undetected		
	The 7 wastewater samples	Undetected	Undetected		
	Known PCR positive swab	Undetected	Undetected		
	No template control	Undetected	Undetected		
	Positive control	30	Undetected		

* Extraction was conducted using QIAamp Viral RNA Mini kit. ** Extraction was conducted using Allprep Power Viral DNA/RNA extraction kit (QIAGEN). *** Extraction was conducted using Allprep Power Viral DNA/RNA extraction kit (QIAGEN) without PM1/B-ME step.

**Table 3 ijerph-19-08515-t003:** SARS-CoV-2 PCR test result of fresh wastewater inlet and aeration samples collected from 9 MBR units at Addis Ababa between 25 December 2021 to 9 January 2022.

Site of Collection	25 December 2021		2 January 2022		9 January 2022	
	Inlet Ct Value	Aeration	Inlet Ct Value	Aeration	Inlet Ct Value	Aeration
Arabsa-01	32.59	NA	29	NA	32	NA
Arabsa-02	Undetected	Undetected	Undetected	Undetected	38	Undetected
Arabsa-03	32	Undetected	31	Undetected	32	Undetected
Tulu Dimtu-01	38	Undetected	31	Undetected	39	Undetected
Tulu Dimtu-02	Undetected	Undetected	34	Undetected	32	Undetected
Tulu Dimtu-03	37	Undetected	30	Undetected	35	Undetected
Oromia-01	32	Undetected	32	Undetected	35	Undetected
Oromia-02	34	Undetected	34	Undetected	34	Undetected
Bulbula	32	Undetected	32	Undetected	35	Undetected
Tap water	Undetected	Undetected	Undetected	Undetected	Undetected	Undetected

## Data Availability

All data related with this research are presented in this manuscript. Furthermore, the Ethiopia Ministry of Health daily COVID-19 report data is available in this link https://covid19.who.int/WHO-COVID-19-global-data.csv (accessed on 12 March 2022).

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
