# Peer review of "Community Wastewater-Based Surveillance Can Be a Cost-Effective Approach to Track COVID-19 Outbreak in Low-Resource Settings: Feasibility Assessment for Ethiopia Context"

_ijerph, 2022, doi:10.3390/ijerph19148515_

Round 1

Reviewer 1 Report

Review of IJERPH-1763382

This manuscript is easy to follow, written well, about the wastewater surveillance in Ethiopia during COVID-19 under the limitation of resources. It

  1. Section 1 and Reference section: These sections are lacking references about wastewater surveillance and waste disposal during COVID-19, such as:
  • Wastewater surveillance systems in COVID-19: Cases in Indonesia, Japan and Viet Nam. Water Science and Technology 83 (2021) 251–256. https://doi.org/10.2166/wst.2020.558
  • COVID-19 waste management in China: Sustainability 14(8) (2022) 4746 https://doi.org/10.3390/su14084746
  • COVID-19 waste management in Indonesia: Sustainability 14(5) (2022) 2556 https://doi.org/10.3390/su14052556
  • COVID-19 waste management in Bangladesh: Case Studies in Chemical and Environmental Engineering 5 (2022) 100177 https://doi.org/10.1016/j.cscee.2021.100177
  • COVID-19 waste management in India: Environmental Science and Pollution Research 28 (2021) 52702-52723 https://doi.org/10.1007/s11356-021-15028-5
  • COVID-19 waste management in Malaysia: Waste Management and Research 39 (2021) 18-26 https://doi.org/10.1177/0734242X20959701

  1. Please use “Justified” style for all the paragraphs in this manuscript.
  2. Please add list of abbreviations in this manuscript.
  3. Please use the proper degree sign °C, not with superscripted zero, superscripted lowercase o, or superscripted uppercase O.
  4. Line 122-143: There are two periods of sample collections, October 2020-December 2020, and December 2021-January 2022. How about January 2021-November 2021? Please provide reasons on why there is no sample collected during that period.
  5. Please write SARS-CoV-2 appropriately, with two dashes, and not “CoV2” (merged and missing a dash), and not “SARS CoV” (missing a dash). Please revise line 25, 25, 29, 33 (keywords), 195, 196, 198, 249, 264, 269, 272, 275, 282, 305, 317, 325, 363, 371, 377.
  6. Please write COVID-19 properly, with a dash. Please revise “COVID 19” without a dash, located at line 33, 58, 211, 220, 257, 270, 272, 276, 355,
  7. Line 131: …MBR is 3 km…--> Please use lowercase k, lowercase m for the abbreviation of the unit kilometer. Because it has a unit, three kilometers can be written as 3 km.
  8. References: Do not use et al, please write the list of authors completely.
  9. References: Please write the name of the journals consistently. For example: “Science of Total Environment” sometimes written in abbreviated form, sometimes in complete form, sometimes with lowercase e.
  10. Reference 7: World Health Organization --> with uppercase H and O.
  11. Reference 22: Please correct “20202”, is it 2020 or 2022?
  12. Reference 23: Please delete this reference from ResearchGate. Please change with proper journal article(s).
  13. Reference 32: Environmental Research --> with uppercase R.
  14. Please separate numbers and their units with a space. For example:

Line 134: Please separate 300 and ml with a space.

Line 136: Please separate 80 and °C with a space.

Line 156: Please separate 30 and ml with a space.

Reviewer 2 Report

The idea of the paper is good and well presented.

The research describes the assessment of the feasibility and utility of a wastewater surveillance system to monitor SARS-CoV-2 RNA in Addis Ababa, Ethiopia, a low-income country. And the results showed that wastewater based SARS CoV2 surveillance approach is feasible for Addis Ababa. COVID-19 wastewater based epidemiological approach can potentially fill the evidence gap in distribution and dynamics of COVID-19 in Ethiopia and other low-income settings. The article is well written, and the text is good. However, it is not something new to the world. It is not a new methodology. If the paper was submitted in December of 2019, at that time it would be a new thing, but not today. This is the problem. Is good, but not something amazing to change the world. It does not add something new to the subject area. It is just “more one paper about COVID in wastewater”. The methodology is ok. The authors need to correct the “terms” such as “ml” change for “mL” (that is the correct form). The same for “µL”. In my opinion, these are reasons to reject the paper. However, if for the journal the novelty is not a problem, the paper is well written. 

Reviewer 3 Report

11) Line 216 -217. Most of the inlet fresh wastewater samples collected from December 25, 2021 to January 09, 2022 and processed within 24 hours of collection were positive for SARS-CoV-2 RNA (Table 3).

It is recommended to indicate the percentage of positive samples.

12)  Line 318-319. The purpose of wastewater aeration process is  to augment microbial growth to remove organic electron donors.

It is recommended to clarify this phrase. The meaning of the process of aeration of wastewater is the sorption of organic contaminants by bacterial microorganisms. How does this relate to "electrons"?

3) Line 446, Ref. 18.

Please, check the authors.

4) Discussion

1.      The cost of examining a single wastewater sample may be higher than the cost of examining a single human nasopharyngeal swab sample. But the study of a single sample of wastewater provides data on a large population of people, and this is very important for epidemiological monitoring and prognosis. In the context of the topic and the title of the article, authors are encouraged to provide information about the cost of wastewater and swab studies from humans and compare them.

1.      

Round 2

Reviewer 1 Report

Review of ijerph-1763382-v2

The authors have addressed the issues raised. The manuscript can be accepted for publication.

Note: Line 342: ...to substantiate.. --> please start "to" with lowercase t, not uppercase T. Thank you.

Reviewer 2 Report

No comments.